# Sequential Learning of Neural Networks for Prequential MDL

**Jorg Bornschein**
bornschein@deepmind.com

**Yazhe Li**
yazhe@deepmind.com

**Marcus Hutter**
mhutter@deepmind.com

## Abstract

Minimum Description Length (MDL) provides a framework and an objective for principled model evaluation. It formalizes Occam's Razor and can be applied to data from non-stationary sources. In the prequential formulation of MDL, the objective is to minimize the cumulative next-step log-loss when sequentially going through the data and using previous observations for parameter estimation. It thus closely resembles a continual- or online-learning problem. In this study, we evaluate approaches for computing prequential description lengths for image classification datasets with neural networks. Considering the computational cost, we find that online-learning with rehearsal has favorable performance compared to the previously widely used block-wise estimation. We propose *forward-calibration* to better align the models predictions with the empirical observations and introduce *replay-streams*, a minibatch incremental training technique to efficiently implement approximate random replay while avoiding large in-memory replay buffers. As a result, we present description lengths for a suite of image classification datasets that improve upon previously reported results by large margins.

## 1 Introduction

Within the field of deep learning, the paradigm of *empirical risk minimization* (ERM, Vapnik (1991)) together with model and hyper-parameter selection based on held-out data is the prevailing training and evaluation protocol. This approach has served the field well, supporting seamless scaling to large model and data set sizes. The core assumptions of ERM are: a) the existence of fixed but unknown distributions $q(x)$ and $q(y|x)$ that represent the data-generating process for the problem under consideration; b) the goal is to obtain a function $\hat{y} = f(x, \theta^*)$ that minimizes some loss $\mathcal{L}(y, \hat{y})$ in expectation over the data drawn from $q$; and c) that we are given a set of (i.i.d.) samples from $q$ to use as training and validation data. Its simplicity and well understood theoretical properties make ERM an attractive framework when developing learning machines.

However, sometimes we wish to employ deep learning techniques in situations where not all these basic assumptions hold. For example, if we do not assume a fixed data-generating distribution $q$ we enter the realm of *continual learning*, *life long learning* or *online learning*. Multiple different terms are used for these scenarios because they operate under different constraints, and because there is some ambiguity about what problem exactly a learning machine is supposed to solve. A recent survey on continual- and life-long learning for example describes multiple, sometimes conflicting desiderata considered in the literature: forward-transfer, backward-transfer, avoiding catastrophic forgetting and maintaining plasticity (Hadsell et al., 2020).

Another situation in which the ERM framework is not necessarily the best approach is when minimizing the expected loss $\mathcal{L}$ is not the only, or maybe not even the primary objective of the learning machine. For example, recently deep-learning techniques have been used to aid structural and causal inference (Vowels et al., 2021). In these cases, we are more interested in model selection or some aspect of the learned parameters $\theta^*$ than the generalization loss $\mathcal{L}$. Here we have little to gain from the generalization bounds provided by the ERM framework and in fact some of its properties can be harmful.

Independent of ERM, compression based approaches to inference and learning such as Minimum Description Length (Rissanen, 1984; Grunwald, 2004), Minimum Message Length (Wallace, 2005)

and Solomonoffs theory of inductive inference (Solomonoff, 1964; Rathmanner & Hutter, 2011) have been extensively studied. In practice, however, they have been primarily applied to small scale problems and with very simple models compared to deep neural networks. These approaches strive to find and train models that can compress the observed data well and rely on the fact that such models have a good chance of generalizing on future data. It is considered a major benefit that these approaches come with a clear objective and have been studied in detail even for concrete sequence of observations; without assuming stationarity or a probabilistic generative process at all (also called the *individual sequence* scenario). The individual sequence setting is problematic for ERM because the fundamental step of creating training and validation splits is not clearly defined. In fact, creating such splits implies making an assumption about what is equally distributed across training and validation data. But creating validation splits can be problematic even with (assumed) stationary distributions, and this becomes more pressing as ML research moves from curated academic benchmark data sets to large user provided or web-scraped data.

In contrast to ERM, compression approaches furthermore include a form of Occam's Razor, i.e. they prefer simpler models (according to some implicit measure of complexity) as long as the simplicity does not harm the model's predictive performance.

The literature considers multiple, subtly different approaches to defining and computing description lengths. Most of them are intractable and known approximations break down for overparameterized model families such as neural networks. One particular approach however, prequential MDL (Dawid & Vovk, 1999; Poland & Hutter, 2005), turns computing the description length $L(\mathcal{D}|M) = -\log p(\mathcal{D}|M)$ into a specific kind of continual or online learning problem: $\log p(\mathcal{D}|M) := \sum_{t=1}^{T} \log p_M(y_t|x_t, \hat{\theta}(\mathcal{D}_{<t}))$, where $\mathcal{D} = \{(x_t, y_t)\}_1^T$ is the sequence of inputs $x_t$ and associated prediction targets $y_t$; $p_M(y|x, \theta)$ denotes a model family $M$ and $\hat{\theta}(\mathcal{D}_{<t})$ an associated parameter estimator given training data $\mathcal{D}_{<t}$. In contrast to ERM which considers the performance of a learning algorithm on held-out data after training on a fixed dataset, prequential MDL evaluates a learner by its generalization performance on $y_t$ after trained on initially short but progressively longer sequences of observations $\mathcal{D}_{<t}$. This resembles an online learning problem where a learner is sequentially exposed to new data $(x_t, y_t)$, however a sequential learner is also allowed to revisit old data $x_{t'}, y_{t'}$ for training and when making a prediction at time $t > t'$. We refer to (Bornschein & Hutter, 2023) for an in-depth discussion of the benefits and challenges when using prequential MDL with deep learning.

**Contributions.** Previous work on computing prequential description lengths with neural networks relied on a block-wise (chunk-incremental) approximation: at some positions $t$ a model is trained from random initialization to convergence on data $\mathcal{D}_{<t}$ and then their prediction losses on the next intervals are combined (Blier & Ollivier, 2018; Bornschein et al., 2020; Jin et al., 2021; Bornschein et al., 2021; Perez et al., 2021; Whitney et al., 2020). We investigate alternatives that are inspired by continual-learning (CL) based methods. In particular, chunk-incremental and mini-batch incremental fine-tuning with rehearsal. Throughout this empirical study, we consider the computational costs associated with these methods.

We propose two new techniques: *forward-calibration* and *replay streams*. *Forward-calibration* improves the results by making the model's predictive uncertainty better match the observed distribution. *Replay streams* makes replay for mini-batch incremental learning more scalable by providing approximate random rehearsal while avoiding large in-memory replay-buffers. We identify exponential moving parameter averages, label-smoothing and weight-standardization as generally useful techniques. As a result we present description lengths for a suite of popular image classification datasets that improve upon the previously reported results (Blier & Ollivier, 2018; Bornschein et al., 2020) by large margins.

The motivation for studying prequential MDL stems from the desire to apply deep-learning techniques in situations violating the assumptions of ERM. In this work however we concentrate on established iid. datasets: stationarity is most challenging for a sequential learner, which has to uniformly learn from all previously examples, and it allows us to evaluate well known architectures and compare their held-out performance against ERM-based training, which has been the focus of the community for decades. We include additional experimental results on non-stationary data in Appendix C.7.3.

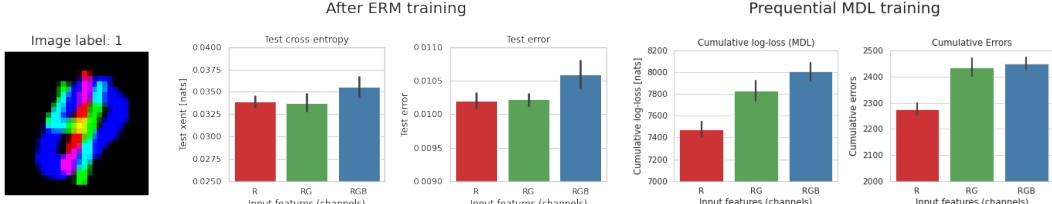

Figure 1: **Left:** Example images. **Center:** Best performing ERM-trained models on held-out data. There is no clear indication which conditioning should be preferred. **Right:** Log-loss and errors accumulated throughout the sequence. We observe that MDL based model selection provides strong evidence that the model conditioned on the R-channel alone should be preferred. In all cases we randomly sampled 50 hyperparameters and use bootstrap-sampling to obtain 95% confidence intervals for the performance of the best performing model. See Figure 2 for detailed regret plots.

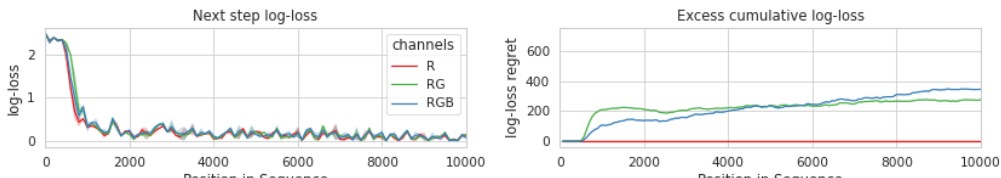

Figure 2: **Left:** Next-step neg. log-loss for the best performing models as a function of the position. The models are trained on all the previous examples. We observe that the models conditioned on the R, R and G, or R, G and B channels all rapidly improve their predictive performance when trained on more data. **Right:** The cumulative neg. log-loss (= description length) relative to the best performing model which is conditioned on the R channel only. During the first $\approx$ 2k examples the models conditioned on irrelevant channels accumulate hundreds of excess nats description length.

**An example: model selection using ERM v.s. MDL.** To showcase the properties of prequential MDL and its bias towards simpler models, we generate images by randomly drawing 3 MNIST examples and superimposing them as red(R), green(G) and blue(B) channels. The label is always associated with the red channel; green and blue are distractors. We train a LeNet (LeCun et al., 1989) convolutional neural network on 100k examples from this dataset; either within the ERM framework by drawing i.i.d. examples, training to convergence and validating on held-out data, or within the MDL framework by treating the data as a sequence, training online with replay buffer (see Section 2 for details) and recording the (cumulative) next-step prediction losses and errors. We compare 3 different models: a model conditioned only on the R channel ($M_R$), on the R and G ($M_{RG}$), or on R, G and B ($M_{RGB}$) channels. Figure 1 (middle) shows the validation-set performance of the best performing models after ERM training. Figure 1 (right) shows description lengths on the same data. It is evident that ERM does not give a clear indication which conditioning should be preferred. However, MDL strongly prefers the model conditioned only on the R-channel. Figure 2 shows a more detailed analysis of the three differently conditioned models in the MDL scenario. We observe that prequential MDL identifies the minimal conditioning because it takes the small-data performance of the models into account. Given sufficient training data the three alternatives perform almost indistinguishable (as seen from their almost parallel regret curves). When we treat the description lengths as evidence for Bayesian model selection to choose between $M_R, M_{RG}$ and $M_{RGB}$ we obtain $p(M_R|\mathcal{D}) = {}^{p(\mathcal{D}|M_R)}\!/\!_{p(\mathcal{D}|M_R) + p(\mathcal{D}|M_{RG}) + p(\mathcal{D}|M_{RGB})}$. With the description length $-\log p(\mathcal{D}|M_R)$ about 300 nats smaller than the ones for $M_{RG}$ and $M_{RGB}$, we have $\approx 1{:}e^{-300}$ odds in favour of $M_R$. The Appendix C.1 contains additional experiments with more architectures and the experimental details; demonstrating that this is a robust result regardless of architectures and optimization hyperparameters.

## 2 ONLINE LEARNING APPROACHES TO PREQUENTIAL MDL ESTIMATION

In this section, we describe practical strategies for computing prequential description lengths with neural networks. Conceptually, computing the cumulative log-loss resembles a continual- or online-learning problem without constraints such as limiting access to previously observed data. In practice, however, some naive approaches such as retraining from scratch to convergence whenever a new

---

**Algorithm 1** Mini-batch Incremental Training with Replay Streams

---

**Require:** data $(x_t, y_t)_{t=1}^T$; augmentation $Aug(.)$; number of replay streams $K$; EMA decay $\alpha$
 1: Initialize: parameters $\theta$; EMA parameters $\bar{\theta} = \theta$; softmax temperature calibration $\beta = 1$;
    Replay positions $\{\rho_k = 1\}_{k=1}^K$; $\mathcal{L}_{preq} = 0$
 2: **for** $t = 1$ **to** $T$ **do**
 3:     Compute next-step loss: $\mathcal{L}_t := -\log p(y_t | x_t, \bar{\theta}, \beta)$
 4:     Update cumulative loss: $\mathcal{L}_{preq} \leftarrow \mathcal{L}_{preq} + \mathcal{L}_t$
 5:     Update temperature calibration: $\beta \leftarrow \beta - \nabla_\beta \log p(y_t | x_t, \bar{\theta}, \beta)$
 6:     Update parameters: $\theta \leftarrow \theta - \nabla_\theta \log p(y_t | \tilde{x}_t, \theta, 1.)$, with $\tilde{x}_t = Aug(x_t)$
 7:     Update EMA parameters: $\bar{\theta} \leftarrow (1 - \alpha)\bar{\theta} + \alpha \theta$
 8:     **for** $k = 1$ **to** $K$ **do**
 9:         Get data from stream $k$: $x \leftarrow x_{\rho_k}$; $y \leftarrow y_{\rho_k}$, advance position $\rho_k \leftarrow \rho_k + 1$
10:         Update parameters: $\theta \leftarrow \theta - \nabla_\theta \log p(y | \tilde{x}, \theta, 1.)$ with $\tilde{x} = Aug(x)$
11:         Update EMA parameters: $\bar{\theta} \leftarrow (1 - \alpha)\bar{\theta} + \alpha \theta$
12:         Maintain replay distribution: with probability $p_{\text{reset}}$ reset $\rho_k \leftarrow 1$ (see Equation (1))
13:     **end for**
14: **end for**
15: **return** $\mathcal{L}_{preq}$

In practice we perform the algorithm with mini-batches instead of individual examples and use gradient based optimizers such as AdamW(Loshchilov & Hutter, 2019) instead of plain SGD.

---

example arrives are infeasible due to their high resource usage. In our study, we consider the compute requirements of each approach by counting the number of floating-point operations (FLOPs) until the learner reaches the end of the data sequence. We do not limit the learner's ability to access or sample previously observed data because storing and reading from large datasets is generally not considered a major technical bottleneck. However, in-memory (RAM) based implementations of replay-buffers for continual learning (rehearsal) suffer from limited capacity because data sets for training deep neural networks often exceed RAM sizes. Below we describe *replay-streams*, a simple approach to approximate random sampling for replay while only utilizing cheap sequential access to data on disk. With this approach, unlimited approximate random rehearsal from data on permanent storage becomes as easy as sampling from large datasets for ERM based optimization.

We compare the following approaches to compute prequential description lengths:

**Chunk Incremental / From-Scratch (CI/FS):** Variants of this approach have been used to compute the recently reported description lengths for deep-learning models (Blier & Ollivier, 2018; Bornschein et al., 2020; Jin et al., 2021; Bornschein et al., 2021; Perez et al., 2021). The data-sequence is partitioned into $K$ non-overlapping intervals; typically of increasing size (i.e. choosing exponentially spaced split points $\{s_k\}_{k=1}^K$, with $s_k \in [2, \dots, n]$, $s_k < s_{k+1}$ and $s_K = N$). For each $k$, a neural network is randomly initialized and trained to convergence on all the examples before the split point $s_k$ and evaluated on the data between $s_k$ and $s_{k+1}$. This corresponds to the block-wise approximation of the area under the curve: $\sum_{i=1}^N \log p(y_i | \hat{\theta}(\mathcal{D}_{<i})) \approx \sum_{k=1}^{K-1} \sum_{j=s_k}^{s_{k+1}} \log p(y_j | \hat{\theta}(\mathcal{D}_{<s_k}))$, where $\hat{\theta}(\mathcal{D})$ denotes parameters after training on data $\mathcal{D}$. To ensure that the model produces calibrated predictions even when $\mathcal{D}_{<s_k}$ is small, we use softmax temperature calibration (Guo et al., 2017): at each stage we split the data $\mathcal{D}_{<s_k}$ into a 90% training and a 10% calibration data. Conceptually, we could perform post-calibration by first training the network to convergence and then, with all parameters frozen, replacing the output layer $\text{softmax}(h)$ with the calibrated output layer $\text{softmax}(\text{softplus}(\beta)h)$, where $\beta$ is a scalar parameter chosen to minimize the loss on calibration data. In practice, we optimize $\beta$ by gradient descent in parallel with the other model parameters. We alternate computing gradient steps for $\theta$, calculated from the training set and using the uncalibrated network (with final layer $\text{softmax}(h)$), with a gradient step on $\beta$, calculated from the validation set using the calibrated network (with final layer $\text{softmax}(\text{softplus}(\beta)h)$). This simple calibration procedure has proven to be surprisingly effective at avoiding overfitting symptoms when training large neural networks on small datasets (Guo et al., 2017; Bornschein et al., 2020).

**Chunk Incremental / Continual Fine-tuning (CI/CF):** Similar to CI/FS, the sequence is split into increasingly larger chunks. But instead of training the model from scratch at each stage, the network

is continuously fine-tuned on the now larger dataset $\mathcal{D}_{<s_k}$. We use the same calibration strategy as for CI/FS. We expect to save compute by avoiding training from scratch. However, recent research suggests that, when first trained on data $\mathcal{D}_A$ and then trained on data $\mathcal{D}_A \cup \mathcal{D}_B$, deep-learning models struggle to converge to a solution that generalizes as well as training on $\mathcal{D}_A \cup \mathcal{D}_B$ from scratch; even when both $\mathcal{D}_A$ and $\mathcal{D}_B$ are sampled from the same underlying distribution. In Ash & Adams (2020) the authors observe that shrinking the model-parameters and adding some noise after training on $\mathcal{D}_A$ can improve the final generalization performance. We therefore run an ablation study and perform *shrink & perturb* operation whenever advancing to the next chunk $k$.

**Mini-batch Incremental / Replay-Buffer (MI/RB)**    This approach uses continual online mini-batch gradient descent with an additional replay buffer to store previously seen examples. At each time $t$, the learner performs a number of learning steps on data in the replay buffer. This could be all or a subset of the data $\mathcal{D}_{<t}$ depending on the capacity of the replay. We propose *forward-calibration* to optimize the calibration parameter $\beta$: each new batch of examples is first used for evaluation, then used to perform a single gradient step to optimize $\beta$, and finally used for training the parameters $\theta$ and placed into the replay-buffer. Calibration is computationally almost free: the forward-pass for the calibration gradient step can be shared with the evaluation forward-pass, and backward-propagation for $\beta$ ends right after the softmax layer. Appendix B.1 contains a more detailed description. If the replay buffer is limited by the memory capacity, we either use a FIFO replacement-strategy or reservoir-sampling (Vitter, 1985) to maintain a fixed maximum buffer size.

**Mini-batch Incremental / Replay-Streams (MI/RS)**    Implementing large replay buffers can be technically challenging: in-memory (RAM) replay buffers have limited capacity and random access and sampling from permanent storage is often slow. We propose a simple yet effective alternative to storing replay data in-memory. We instead assume that the data is stored in its original order on permanent storage and we maintain $K$ replay streams (pointers into the sequence) , each reading the data in-order from the sequence $(x_1, y_1) \ldots (x_T, y_T)$. We can think of a replay-stream as a file-handle, currently at position $\rho \in 1 \ldots T$. A read operation yields $(x_\rho, y_\rho)$ and increments the position to $\rho \leftarrow \rho + 1$. We denote the position of the $k$th replay-stream with $\rho_k$. Every time the learner steps forward from time $t$ to $\tilde{t} = t+1$ we also read a replay example from each of the $K$ replay-streams and perform gradient steps on them. By stochastically resetting individual streams with probability $p_{\text{reset}}$ to position $\rho \leftarrow 1$, we can influence the distribution of positions $\rho$. If we wish to use uniform replay of previous examples we reset with probability $p_{\text{reset}} = 1/\tilde{t}$. If instead we wish to prioritize recent examples and replay them according to $p_{\text{replay}}(a)$, where $a = \tilde{t} - i$ is the age of the example $(x_i, y_i)$ relative to the current position of the learner $\tilde{t}$, we use

$$p_{\text{reset}}(\tilde{t}) = \frac{\sum_{a=t}^{\tilde{t}} p_{\text{replay}}(a)}{\sum_{a'=0}^{\tilde{t}} p_{\text{replay}}(a')} \tag{1}$$

Note that $p_{\text{replay}}$ can be an unnormalized distribution and that by setting $p_{\text{replay}} \propto 1$ we recover $p_{\text{reset}} = 1/\tilde{t}$ for uniform replay. See Algorithm 1 for details.

## 3    EXPERIMENTS

We empirically evaluate the approaches from Section 2 on a suite of models and image classification data sets. We use MNIST (LeCun et al., 2010), EMNIST, CIFAR-10, CIFAR-100 (Krizhevsky, 2009) and ImageNet (Russakovsky et al., 2015) and randomly shuffle each into a fixed sequence of examples. With MDL, we do not require validation or test set and could merge it with the training data. But because the data is known to be i.i.d. and we compare results with established ERM-based training, we instead maintain the standard test sets as held-out data. It must be emphasized that the primary objective throughout this study is to minimize the description length. All hyperparameters are tuned towards that goal. When we report test set performance it should be considered a sanity check. For more detailed analyses we will plot the "*regret*" a method $M_i$ accumulates relative to some comparator method $M_c$: $R_i(t) := -\log p(\mathcal{D}_{\le t}|M_i) + \log p(\mathcal{D}_{\le t}|M_c)$. We evaluate the following architectures: MLPs, VGG (Simonyan & Zisserman, 2015), ResNets (He et al., 2016), WideResNets (Zagoruyko & Komodakis, 2016) and the transformer-based DeiT (Touvron et al., 2021) architecture. We refer to a variant of the VGG architecture with weight-standardized convolutions (Qiao et al., 2019) and with batch normalization (Ioffe & Szegedy, 2015) as *VGG++*.

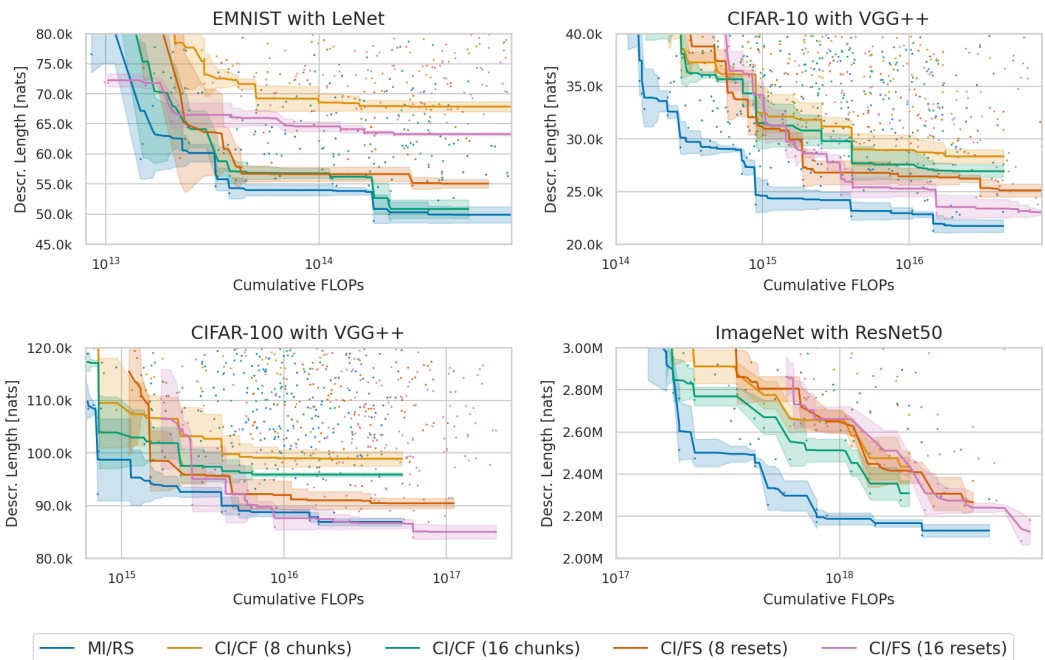

Figure 3: Shortest description lengths as function of compute resources consumed for different methods. For each method (in different colors) we run 250 (100 for ImageNet) independent experiments with randomly sampled hyperparameters. Each dot represents such an experiment and the solid lines show the Pareto-front of the best archived performance for different compute budgets. With the exception of CIFAR-100 in the large-FLOPs regime, the mini-batch incremental appoach MI/RS results in the shortest description lengths.

For the bulk of our experiments, we use the AdamW (Loshchilov & Hutter, 2019) as optimizer; a series of control experiments however suggest that the quantitative results generalize to optimizers such as RMSProp and simple mini-batch SGD with momentum. We identified the following techniques that consistently improve the results and use them throughout this paper if not mentioned otherwise:

**Exponentially moving average (EMA)** Similar to Polyak averaging (Polyak & Juditsky, 1992), EMA maintains a moving average of the model parameters $\theta$ and is used for next-step evaluations as well as when tuning the softmax temperature $\beta$.

**Label smoothing** (Szegedy et al., 2015) A reliable regularizer, label smoothing often helps for the continual fine-tuning methods such as MI/RB, MI/RS and CI/CF.

**Weight standardization** For the VGG architecture, weight standardized convolutions lead to significant improvements - bigger than for conventional ERM-based training. For other architectures such as ResNets, the results are not conclusive.

## 3.1 COMPARISON OF ONLINE LEARNING APPROACHES.

To compare the effectiveness of the different approaches listed in Section 2, we run a broad random hyperparameter sweep for EMNIST with LeNet, CIFAR-10 and CIFAR-100 with VGG++ and ImageNet with ResNet50. The hyperparameter intervals depend on the data and are detailed in Appendix B. We sample learning rate, EMA step size, batch size, weight decay; but crucially also number of epochs (or, correspondingly, number of replay streams for MI/RS) and an overall scaling of the model width (number of channels). We thus cover a large range of computational budgets, spanning about 2 orders of magnitude. When the capacity of the replay-buffer matches the total length of the sequence, MI/RB and MI/RS are equivalent. In these cases we omit MI/RB and instead use MI/RS to represent online-learning with unlimited rehearsal. A study of replay buffer v.s. replay streams can be found in Section 3.1. We show the overall Pareto front for these experiments in Figure 3. Mini-batch incremental learning obtains the lowest description lengths for a wide range

| Data | Model | MI/RS | | | | | ERM | | |
|---|---|---|---|---|---|---|---|---|---|
| | | Cumulative | | Test | | | Test | | |
| | | Loss | Errors | Loss | Error(%) | FLOPs | Loss | Error(%) | FLOPs |
| MNIST | LeNet | 4.4k | 1362 | 0.03 | 1.0 | 1.4e13 | 0.03 | 0.8 | 1.5e13 |
| MNIST | MnistNet | 2.1k | 643 | 0.02 | 0.5 | 4.5e14 | 0.02 | 0.5 | 3.2e14 |
| CIFAR-10 | VGG++ | 22.1k | 6.6k | 0.24 | 5.8 | 5.9e15 | 0.21 | 6.9 | 8.0e15 |
| CIFAR-10 | WRN-28-10 | 22.9k | 7.5k | 0.23 | 7.4 | 3.7e16 | 0.20 | 6.7 | 5.4e16 |
| CIFAR-100 | VGG++ | 93.7k | 23.1k | 1.14 | 32.0 | 4.2e15 | 1.14 | 28.1 | 2.6e15 |
| CIFAR-100 | WRN-28-10 | 87.0k | 22.3k | 1.10 | 30.3 | 2.5e16 | 1.08 | 28.3 | 4.3e16 |
| ImageNet | VGG++ | 2.55M | 549k | 1.63 | 36.8 | 7.5e18 | | | |
| ImageNet | ResNet-34 | 2.35M | 507k | 1.30 | 30.9 | 1.6e18 | 1.25 | 29.9 | 7.0e17 |
| ImageNet | ResNet-50 | 2.21M | 496k | 1.29 | 29.1 | 8.5e17 | 1.15 | 28.7 | 1.9e18 |
| ImageNet | ResNet-50[†] | 1.93M | 431k | 1.06 | 26.1 | 4.0e18 | | | |
| ImageNet | ResNet-101 | 1.97M | 447k | 1.20 | 28.2 | 3.1e18 | 1.10 | 26.6 | 4.6e18 |
| ImageNet | DeiT-S | 2.73M | 588k | 1.40 | 31.7 | 1.2e18 | 1.22 | 28.1 | 4.8e18 |
| *Prior Works:* | | | | | | | | | |
| MNIST(Blier & Ollivier, 2018) | MnistNet | 2.8k | | | 0.5 | | | | |
| CIFAR-10(Blier & Ollivier, 2018) | ≈ VGG++ | 31.4k | | | 6.7 | | | | |
| ImageNet(Bornschein et al., 2020) | ResNet-50 | 3.32M | | | - | | | | |

Table 1: **Left half of the table**: Suite of results when applying MI/RS for description length estimation. Cummulative loss in nats; cumulative error counts the number of prediction errors after reaching the end of the sequence. The test set metrics report the performance of the model selected by lowest description length. **Right half**: Results when using the same architecture and hyperparameter-sweep for conventional ERM training. [†] Manually tuned hyper parameters.

of computational budgets. Only for very large compute budgets on CIFAR-100 we observe that the block-wise approximation with training from scratch (CI/FS) has a better results.

**MDL for Deep Neural Networks.** We apply MI/RS to a wider range of architectures. For most experiments we again use random hyperparameter sweeps, this time however without scaling the size of the models and with an increased lower bound for the number of replay streams (see Appendix B). We present our results for the best run in each sweep and contrast them with previously reported description lengths from the literature in Table 1. We also show the test set performance of these models after they reached the end of training data. The rightmost columns show the test set performance when performing the same hyperparameter sweep for conventional ERM training on the full data set. Note that these runs do not use any kind of learning rate annealing, which is otherwise often employed to improve results.

Based on these experiments we plot a regret comparison for different architectures trained on ImageNet in Figure 4 (left). We observe that the additional depth and capacity of a ResNet101 is beneficial from the very beginning of the data-sequence. Figure 4 (right) shows the effect of scaling the number of channels of a VGG++ architecture on CIFAR-100 and it is evident that decreasing the model width has negative effects throughout the sequence and not just when the data gets larger.

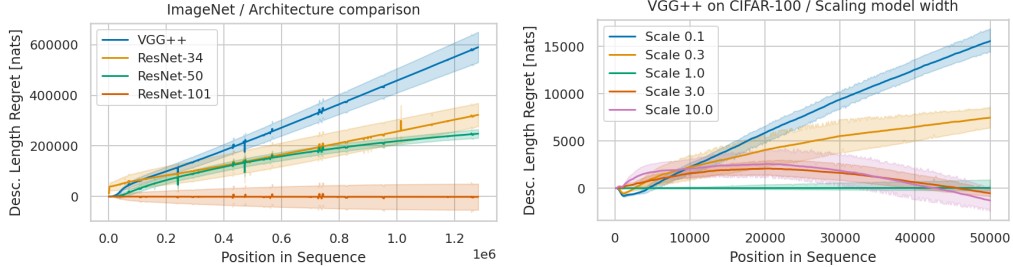

Figure 4: **Left**: Comparing architectures on ImageNet. **Right**: Scaling the size of VGG++.

**Replay Buffer and Replay Streams** are equivalent for short data sequences, they differ when the length of a sequence exceeds the capacity of the replay buffer. In this section, we study the effects of limiting the replay buffer size. Figure 5 shows that reducing the capacity has a significant negative effect on the results. In addition, the strategy of which samples to keep in the buffer plays an important role. Using reservoir sampling instead of a FIFO replacement policy leads to further severely degraded results. This is maybe not surprising because with reservoir-sampling insertion to the buffer is stochastic; a fraction of the examples are thus never replayed at all. Appendix C.5 contains ablations for the number of replay-streams on CIFAR-100 and ImageNet.

**Ablations.** We provide ablations for using forward-calibration, label smoothing and weight standardization in Table 2. It is evident that each of these techniques helps obtaining better description lengths for the datasets and architectures under consideration. Regret plots for these ablations can be found in Appendix C.

| Data | Model | None | +FC | +LS | +WS | +ALL |
|------|-------|------|-----|-----|-----|------|
| CIFAR-10 | VGG++ | 32.4k | 31.8k (-0.6k) | 32.1k (-0.3k) | 26.4k (-5.9k) | 23.3K (-9.0K) |
| CIFAR-100 | VGG++ | 103.4k | 100.7k (-2.7k) | 102.8k (-0.6k) | 93.1k (-10.2k) | 89.1k (-14.2) |
| ImageNet | ResNet50 | 2.33M | 2.27M (-60.0k) | 2.28M (-50k) | | 2.21M (-120k) |

Table 2: Description lengths (in nats) with various techniques added to the baseline model: forward-calibration (FC), label smoothing (LS) and weight standardization (WS).

**Plasticity and the Warm-starting problem** We take a closer look at the large-FLOPs regime for CIFAR-100 where CI/FS obtained better results than MI/RS. Figure 6 (left) shows the regret comparison between the best runs for each training method. The split positions $s_k$ are clearly visible as kinks in the graph for CI/FS and CI/CF. For CI/CF the plot shows that right after fine-tuning, the predictive performance is approximately equal to MI/RS. For CI/FS we see that the regret relative to MI/RS decreases: it has better predictive performance. This effect persists even after running extensive additional hyperparameter sweeps. We conjecture that this is a signature of the warm-starting problem described in (Ash & Adams, 2020). Figure 6 (right) shows the Pareto front for the final test set accuracy of the best models (selected by their description length) over FLOPs. We observe a large gap between fine-tuning based methods and CI/FS; which is consistent with this hypothesis. Note that the last split-point $s_K = T$ implies that these models are effectively trained like in a conventional ERM paradigm. Appendix C shows test set pareto fronts for a range of datasets. Close inspection of various regret plots sometimes reveals the same signatures, suggesting that the effect is ubiquitous, but often much less pronounced. It is plausible that this also contributes to the gap in final test set performance in Table 1.

## 4 DISCUSSION & RELATED WORK

Compression based approaches to inference and learning, such as Solomonoff induction (Solomonoff, 1964), Minimum Message Length (Wallace, 2005) and Minimum Description Length (Rissanen, 1984; Grunwald, 2004), have been extensively studied. The Bayesian and variational Bayesian approaches to estimating code lengths have been popular and, at least outside the field of deep learning, also been very successful. This line of work includes methods such as AIC and BIC, as well as more sophisticated approximations to the posterior distribution over model parameters for many kinds of models, including neural networks (Hinton & Van Camp, 1993; MacKay, 2003). The description length depends crucially on the quality of these posterior approximations. Unfortunately, estimating the parameter posterior for modern neural networks is notoriously hard and an area of active research (Izmailov et al., 2021). Blier & Ollivier (2018) demonstrated that much shorter code lengths can be achieved by using the prequential approach with standard neural network architectures and optimizers. The prequential perspective with the block-wise approximation technique have been

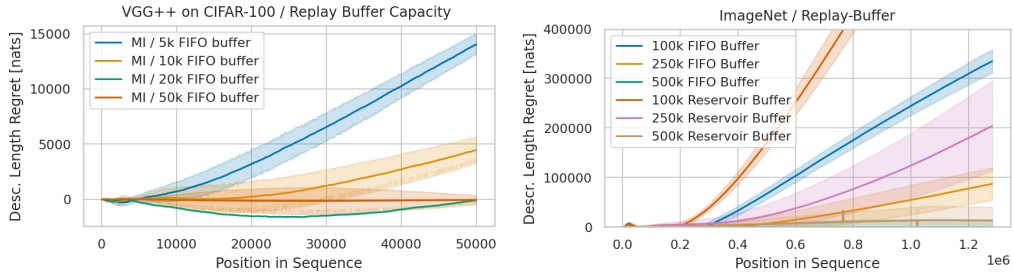

Figure 5: The effect of reducing the replay-buffer capacity. **Left**: Regret for different replay buffer sizes for CIFAR-100. **Right**: Comparing buffer capacity and FIFO vs. reservoir sampling based replacement strategies for ImageNet with ResNet50.

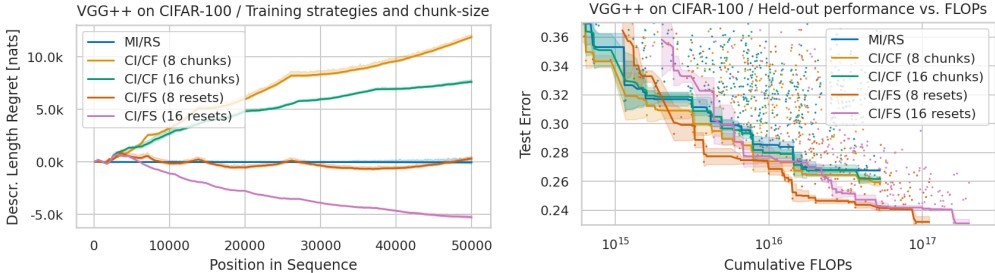

Figure 6: **Left**: Comparing the regret for CI/FS and CI/CF relative to MI/RS for CIFAR-100. **Right**: Pareto-front for the test set performance after reaching the end of the sequence

successfully used in the context of image-classification (Bornschein et al., 2020), natural language processing (Voita & Titov, 2020; Jin et al., 2021; Perez et al., 2021) and causal structure discovery (Bornschein et al., 2021).

With our work, we aim to simplify and improve prequential code length estimation for neural networks with insights from the field of continuous learning (CL). For CL, fine-tuning with rehearsal is considered a strong baseline and large number of methods have been proposed to maintain its benefits while minimizing the size of the replay buffer (Mnih et al., 2013; Delange et al., 2021). We instead propose a method that makes it practical and relatively cheap to work with full rehearsal. Caccia et al. (2022) investigate very similar scenario to ours; largely phrased around how long a learner should wait and accumulate data when it is compute constraint; most experiments however consider the case without full rehearsal. He and Lin (He & Lin, 2020) discuss the close relationship between prequential coding and CL, however focus on compressive replay instead of accessing previous data. In general, the literature on CL methods and evaluation metrics is too broad to be adequately discussed here (Hadsell et al., 2020). However, we would not be surprised if some existing CL methods have the potential to vastly improve prequential MDL computation. In this sense, we see our work as a step of bringing CL and prequential MDL related works closer together.

Besides of taking inspiration from CL to improve description lengths, we believe that the MDL perspective can help guide future CL research: Much of the CL literature is rooted in an ERM-based perspective and thus considers distinct training, validation and test sets at different stages during learning. This has potentially played a role in the proliferation of many different evaluation metrics and training protocols used in CL research (Hadsell et al., 2020). This heterogeneity of the evaluation metrics makes it challenging to compare results. Forward-transfer is a commonly used evaluation metric and it is closely related to a prequential evaluation. However, it is often computed on separate held-out sets after training and does not take sample-efficiency into account as prequential MDL does. Recent work has embraced forward-validation in terms of average prediction-error on future training data as a evaluation metric, thus bringing it closer to a prequential MDL based evaluation but without discussing the connection (Cai et al., 2021; Lin et al., 2021). MDL has been conceived and analyzed as a theoretically well motivated objective when dealing with non-stationary data; without reference to the practical difficulties of actually performing parameter estimates. It is a challenging objective even for learners without constraints on compute and (replay-) memory.

## 5 CONCLUSIONS & LIMITATIONS

In this paper, we compare different approaches to computing prequential description lengths while considering their computational cost. We find that continual learning inspired mini-batch incremental training with full rehearsal is the preferred method over a wide range of computational budgets. We introduce two techniques that either improve results, or make it practical to apply full rehearsal even for very large data. As a result we report description lengths for a range of standard data sets that improve significantly over previously reported figures. Some of our results show that the warmstarting-effect can have a negative effect on the performance of continual fine-tuning. We also generally observe a gap in the final held-out performance after training models within the prequential framework vs. training models with the ERM-based approach. We are not aware of any technique to reliably mitigate the impact. It would be an important topic to tackle for future works. Furthermore,

learning rate schedules such as cosine decay are popular and effective techniques for achieving better performance in ERM-based training. It is not obvious how to best translate them to the prequential learning scenario as the learner constantly receives new data.

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

SEQUENTIAL LEARNING OF NEURAL NETWORKS FOR PREQUENTIAL MDL
SUPPLEMENTARY MATERIAL

## A  PRELIMINARIES

Throughout the main paper and the appendix we use the following notation:

| Symbol | Description | Example |
|---|---|---|
| $x_{1\ldots T}$ | Ordered sequence of observations; potentially multivariate | Images |
| $y_{1\ldots T}$ | Ordered sequence of prediction targets | Classification labels |
| $\mathcal{D}$ | Observed data for training | $\mathcal{D} = (x_{1\ldots T}, x_{1\ldots T})$ |
| $\mathcal{D}_{<t}$ | Observed data up to $t-1$ | $\mathcal{D} = (x_{1\ldots t-1}, x_{1\ldots t-1})$ |
| $p_M(Y|X, \theta)$ | Parametric model $M$ | Neural network with softmax |
| $p_M(Y|X, \theta, \beta)$ | Parametric model $M$ with softmax temperature $\beta$ | See (Guo et al., 2017) |
| $\hat{\theta}(\mathcal{D})$ | Estimated model parameters after training on $\mathcal{D}$ | Result of SGD training |
| $\hat{\theta}^{\text{MLE}}(\mathcal{D})$ | Maximum likelihood solution $\arg\max_\theta \prod_t p_M(y_t|x_t, \theta)$ | |
| $\{\rho_k = 1\}_{k=1}^K$ | Position of $K$ independent data readers for replay-stream training | See Algorithm 1 |

## B  EXPERIMENTAL DETAILS

### B.1  FORWARD CALIBRATION

We train the neural network with standard cross-entropy loss and the categorical prediction-head $p = \text{softmax}(h)$, where $h$ is a vector of logits generated by the neural network. When using the model for evaluation on next-step data or on test set examples, we replace it with a calibrated prediction head $p = \text{softmax}(\text{softplus}(\beta)\, h)$, where $\beta$ is a scalar temperature parameter (Guo et al., 2017). We use *forward-calibration* to optimize the calibration parameter $\beta$: each new batch of examples from the training sequence is used first for evaluation, then used to perform a single gradient step to optimize $\beta$, and finally used for a training to optimizer $\theta$ and placed into the replay-buffer. At the beginning of training we initialize $\beta$ to $\beta_0 = \log(\exp(1) - 1) \approx 0.5416$, such that $\text{softplus}(\beta_0) = 1$. We use the same optimizer as for the model parameters $\theta$, however scale the learning rate by a factor of $\sqrt{K}$, where $K$ is the number of training steps per evaluation step forward (i.e., number replay-streams plus one). Calibration is computationally almost free: the forward-pass for the calibration gradient step can be shared with the evaluation forward-pass, and backward-propagation for $\beta$ ends shortly after the softmax layer.

### B.2  MNIST, EMNIST AND RGB-MNIST HYPERPARAMETERS

We run experiments with various model architectures: a) MLP with 1, 2 or 3 hidden layers, 512 units each, with dropout, and ReLU non-linearities, b) LeNet (LeCun et al., 1989) and c) the VGG inspired architecture used by (Blier & Ollivier, 2018), which we call MnistNet here.

We use the same hyperparameter and sampling intervals for all three model architectures. For the pareto front experiment we extend the number of replay-streams range to $10\ldots 100$; otherwise we use the same intervals.

| Parameter | Distribution | Values / Interval |
|---|---|---|
| Number of replay-streams | log-uniform | $25 \ldots 100$ |
| Learning rate | log-uniform | 1e-4 $\ldots$ 3e-3 |
| AdamW $\epsilon$ | log-uniform | 1e-4 $\ldots$ 1 |
| EMA step size (Polyak averaging) | log-uniform | 1e-3 $\ldots$ 1e-1 |
| Weight decay | log-uniform | 1e-4 $\ldots$ 1. |
| Batch size | fixed | 32 |
| Label smoothing | fixed | 0.001 |

## B.3 CIFAR-10 AND CIFAR-100 WITH VGG++, AND WIDERESNET HYPERPARAMETERS

We use the same hyperparmater sampling intervals for all model architectures. Following Zagoruyko & Komodakis (2016) we use only minimal augmentations during training: Images will be horizontally flipped with a probability of 0.5. For the pareto-front plots we extend the *number of epochs* (= number of replay-streams) range to $10 \ldots 100$ and

| Parameter | Distribution | Values / Interval |
|---|---|---|
| Number of replay-streams | log-uniform | $25 \ldots 100$ |
| Learning rate | log-uniform | 1e-4 $\ldots$ 3e-3 |
| AdamW $\epsilon$ | log-uniform | 1e-4 $\ldots$ 1 |
| EMA step size (Polyak averaging) | log-uniform | 1e-3 $\ldots$ 1e-1 |
| Weight decay | log-uniform | 1e-4 $\ldots$ 1e-1 |
| Batch size | uniform | {32, 64, 128} |
| Label smoothing | fixed | 0.01 |

## B.4 IMAGENET WITH VGG++ AND RESNETS HYPERPARAMETERS

We use randaugment for data augmentation and the same hyperparameter intervals for all experiments. For Pareto front experiments we additionally scale the architecture size (number of channels throughout) from $1/4\times$ to $4\times$ and extend the number of replay-streams interval to $10 \ldots 100$.

| Parameter | Distribution | Values / Interval |
|---|---|---|
| Number of replay-streams | log-uniform | $25 \ldots 100$ |
| Learning rate | log-uniform | 1e-4 $\ldots$ 3e-3 |
| AdamW $\epsilon$ | log-uniform | 1e-4 $\ldots$ 1 |
| EMA step size (Polyak averaging) | log-uniform | 1e-3 $\ldots$ 1e-2 |
| Weight decay | log-uniform | 1e-4 $\ldots$ 1. |
| Batch size | fixed | 512 |
| Label smoothing | fixed | 0.01 |

We report additional baseline metrics for the standard ResNet-50 model on ImageNet, obtained by running the same hyperparameter-sweep:

- Replay Streams: pMDL score 2.21M

- Replay Buffer (500k capacity): pMDL score 2.26M nats

- Chunkwise-from-scratch (16 split points, <1e19 FLOPs): pMDL score 2.52M nats,

Note that for chunkwise-from-scratch we use the techniques from Bornschein et al. (2020), who report 3.32M nats. Using better (wider) hyperparameter search and using more split-points we thus found a significantly better result for their method.

Using a 4x wider architecture:

- Replay Streams: pMDL score 2.18M nats

- Replay Buffer (500k capacity): pMDL score 2.22M nats

- Chunkwise-from-scratch (16 split points, 1e19 FLOPs): pMDL score 2.18M nats,

## B.5 CLOC WITH RESNETS HYPERPARAMETERS

We use randaugment for data augmentation and the same hyperparameter intervals for all experiments.

| Parameter | Distribution | Values / Interval |
|---|---|---|
| Number of replay-streams | log-uniform | $10 \ldots 25$ |
| Learning rate | log-uniform | 1e-4 $\ldots$ 3e-3 |
| EMA step size (Polyak averaging) | log-uniform | 1e-3 $\ldots$ 1e-2 |
| Batch size | fixed | 128 |
| Label smoothing | fixed | 0.1 |

## C  ADDITIONAL EMPIRICAL RESULTS

### C.1  RGB-MNIST FEATURE SELECTION

To illustrate that the model-selection property demonstrated in section 1 is a robust property of (prequential) MDL, we here present a suite of results on 3 different model architectures: 1) am MLP with 2 hidden layers, 512 units each; 2) the LeNet architecture from (LeCun et al., 1989); and 3) the much higher capacity and better tuned convnet from (Blier & Ollivier, 2018).

In all cases we run the same hyperprameter sweep detailed in Appendix B. We show the regret plot for the best performing model in each sweep relative to the model conditioned only on the R channel. We see pMDL reliably and with confidence determines that the model conditioned on R only is the most appropriate one for this data; and we see that the small-data performance at the beginning of the sequence is crucial as the relative generalization performance becomes more similar as the length of the sequence increases:

**MLP Regret Plot**

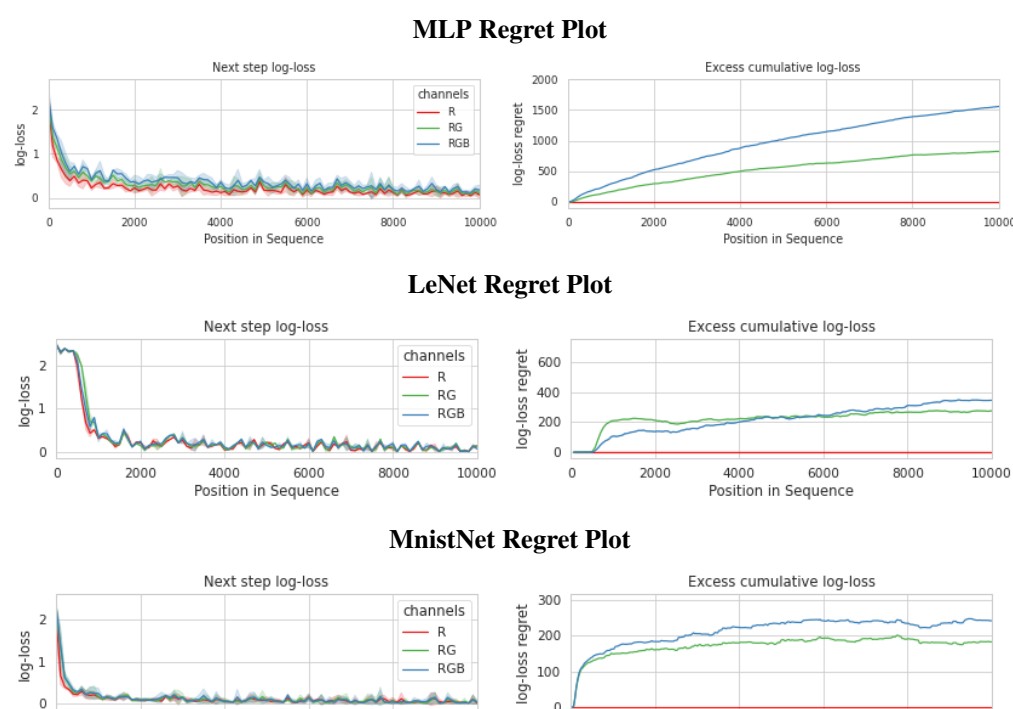

**LeNet Regret Plot**

**MnistNet Regret Plot**

## C.2 Pareto Fronts

**Left:** Description lengths as function of compute resources consumed for a selection of datasets and models. **Right:** Test set performance after reaching the end of the training data. We generally observe a gap between models that train from scratch (CI/FS) and continual finetuning methods. The gap is very pronounced for CIFAR-100. We do not observe a systematic gap between MI/RS and CI/CF.

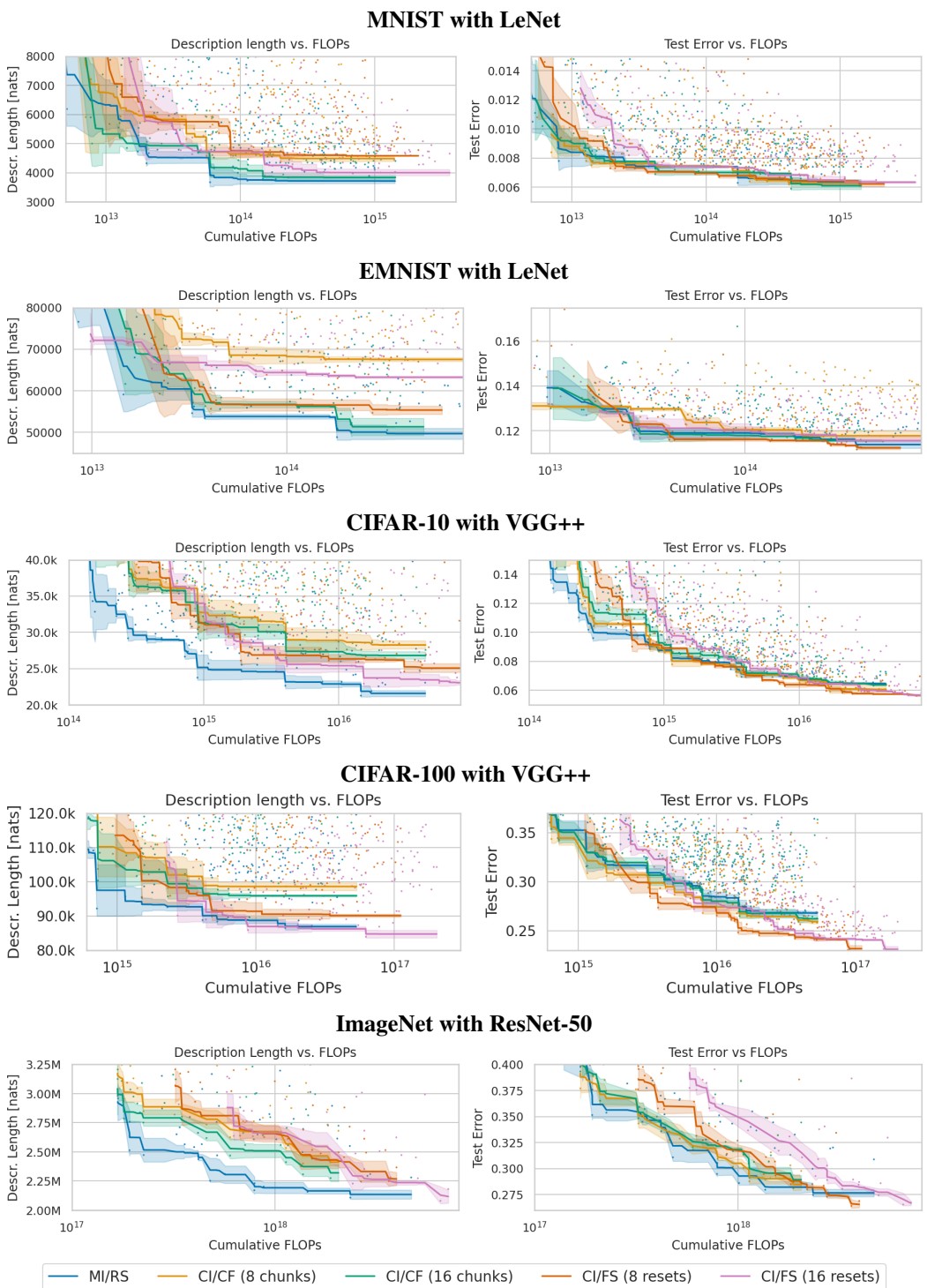

## C.3 MODEL DEPTH ABLATION

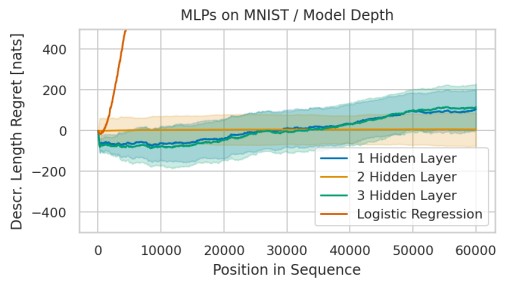 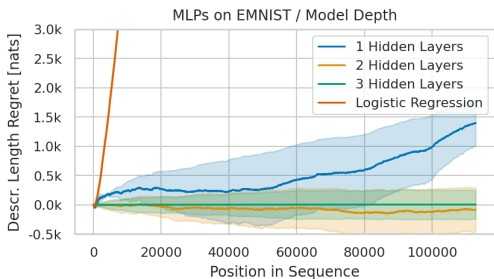

## C.4 WEIGHT-STANDARDIZATION ABLATION

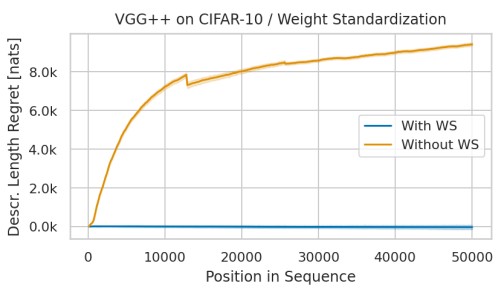 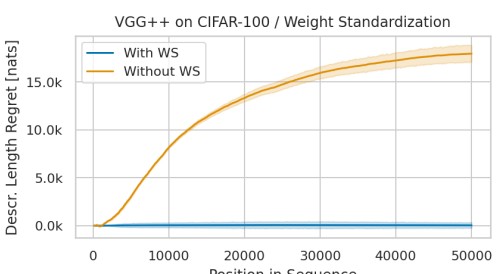

## C.5 NUMBER OF REPLAY STREAMS

We run ablations for different number of replay-streams ($K$ in Algorithm 1) on CIFAR-100 and ImageNet. The results on ImageNet show that performing too much replay (choosing K too large) can be harmful.

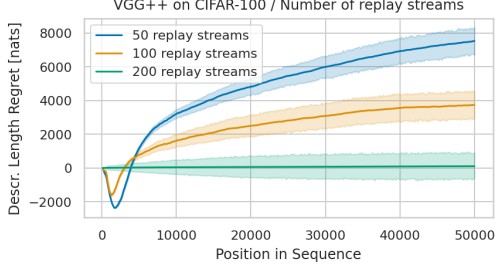 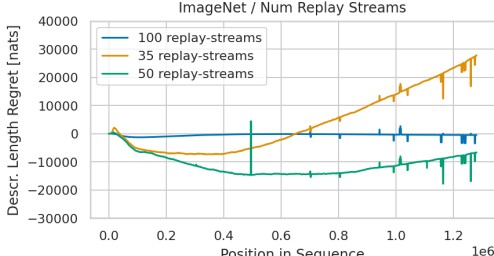

## C.6 SCALE AND SHAPE OF THE REPLAY DISTRIBUTION

Besides of uniform replay for replay-stream training we experimented with alternative replay distributions. Exponentially decaying replay ($p_{\text{replay}}(a) = \lambda \exp(-\lambda a)$) is popular in continual-learning with reservoir sampled replay buffers, but lead to significant longer description lengths and suboptimal results in our experiments.

As an compromise between uniform replay and exponential decay we experimented with long-tailed Pareto distributions: $p_{\text{replay}}(a) = \frac{\alpha}{(a/\lambda)^{(\alpha+1)}}$. The scale parameter $\lambda$ determines to what extend old examples are replayed and we use a fixed shape parameter $\alpha = \log_4 5 \approx 1.16$ for all out experiments:

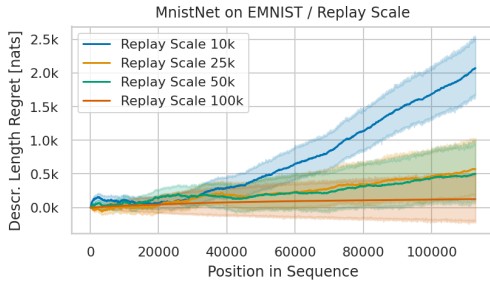 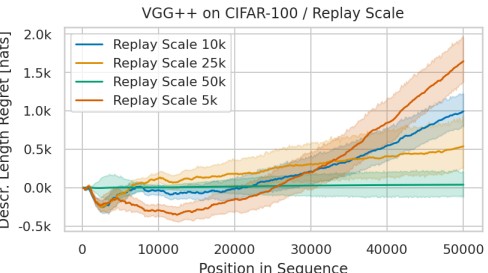

Our experiments suggest that flat, almost uniform replay distributions lead to the shortest description lengths for the sequences from i.i.d. sources. On non-stationary sequences however, long-tailed replay that favours recent data, but maintains finite probability of replaying even the earliest examples works best.

## C.7 NON-STATIONARY SEQUENCES

In this study we concentrate on data sequences from stationary (i.i.d) data sources to ensure we can compare our results with established ERM based training on heldout-data, and, because stationary data is arguably one of the most challenging scenarios for a sequential learner: there is no benefit in focusing on recently observed data and instead a globally optimal predictive model must be learned.

Nevertheless, one major benefit of MDL over ERM is that it is well defined for individual and non-stationary sequences. We therefore present a series of experiments on non-stationary data. We leave it to future work to analyze and to improve online learners on non-stationary data more thoroughly.

### C.7.1 SPLIT CIFAR-10

We derive a non-stationary sequence from the CIFAR-10 dataset by treating it as 5 disjunct binary classification tasks between the CIFAR classes 0 vs. 1 (Task-A), 2 vs. 3 (Task-B), ..., 9 vs 10 (Task-E). We train the model sequentially on examples from each task, however instead of evaluating *forgetting* on held-out data after each new task, which is the standard protocol in continual learaning, we here insert new examples from previous tasks into the non-stationary sequence. A learner can therefore demonstrate good performance by rapidly re-adapting to a previous task from a few examples. The problem is task agnostic because we do not supply information about the current task-id to the learner. We concatenate unique examples from the tasks into a joint sequence: [0, 6000): Task-A, [6000, 13000): Task-B, [13000, 14000): Task-A, [14000, 22000): Task-C, [22000, 23000): Task-A, [23000, 24000): Task-B, [24000, 33000): Task-D, [33000, 34000): Task-A, [34000, 35000): Task-B, [35000, 36000): Task-C [36000, 46000): Task-E, [46000, 47000): Task-A, [47000, 48000): Task-B, [48000, 49000): Task-C, [49000, 50000): Task-D

We run the broad random hyperparameter sweeps for the VGG++ model, however add the scale of the pareto-distributed replay distribution and the capacity of the replay buffer as an additional random hyperparameter between $[1000 \cdots 50000]$. We show the next-step and cumulative next-step performance for the best performing replay-stream and replay buffer learners:

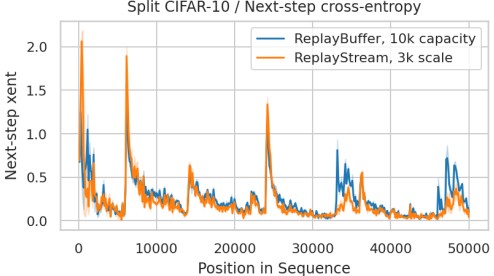 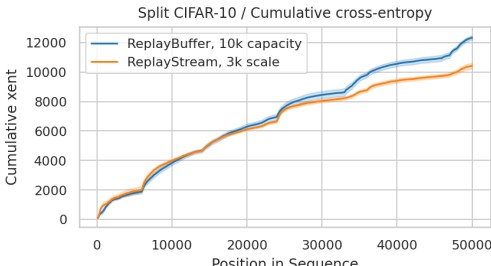

The regret plot for the same experiments show the difference more clearly. We also show the regret for the best replay-stream model with various techniques disabled:

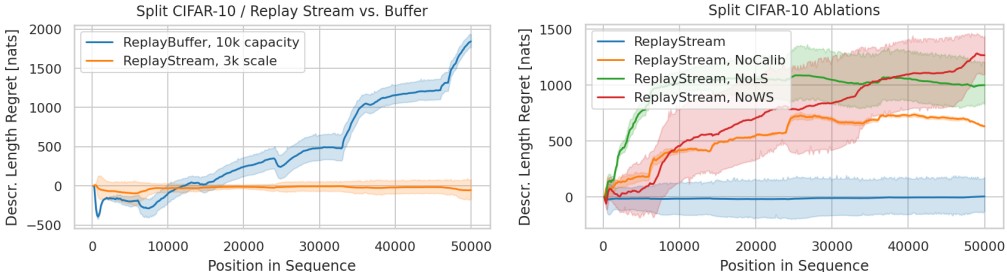

Like in the previous plots, the shaded areas indicate the 99% confidence intervals obtained from bootstrap sampling the experiments in the hyperparameter sweep. We see that multiple replay-stream learners have significantly better performance than the best replay-buffer learner. Given that this dataset is small and fits completely into RAM, we can close the gap between replay-streams and replay-buffers by configuring a replay-buffer with 50k capacity and using recency-biased sampling; which is the only difference between these learners.

Finally, we run an ablation study with 3 different scales for pareto-distributed replay; and the regret plot for 3 replay-buffer learners with 3 different capacities of the uniform sampled replay-buffer. The results were obtained by running the same hyperparameter sweeps as before, however with the replay distribution or replay-capacity fixed to 1k, 3k or 10k respectively. We observe that the width of the replay distribution and the capacity of the replay-buffer trade-off between faster adopting to completely new tasks, and having better performance on recurring tasks:

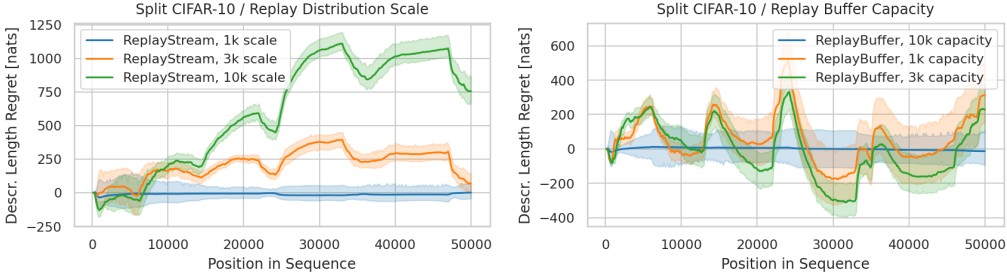

### C.7.2   VEHICLE CLASSIFICATION DATA

We run experiments on the vehicle classification data sequence introduced by Duarte & Hu (2004). The sequence consists of about 80k observations with 100 features each; the classification task is to predict the vehicle category. We use a 2 hidden-layer MLP as the model architecture with the same hyperparamter-sweep from the VGG++ experiments. We show the next-step performance, the corresponding regret plots for replay-stream vs replay-buffer training, and regret plots with label smoothing or forward calibration disabled.

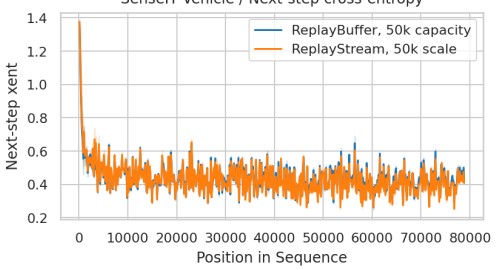
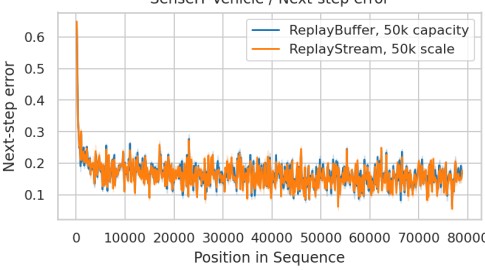

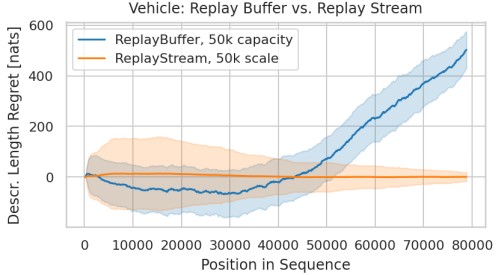 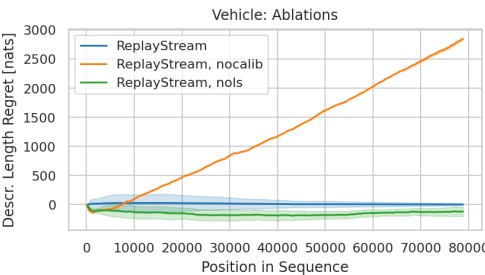

Over the whole sequence we obtain an average forward validation error of $15.5\%$ ($84.5\%$ accuracy). This is competitive with the more complex and ensemble-based methods described by Celik et al. (2022) ($85\%$ accuracy), and an significant improvement over the baseline methods described therein ($\approx 82\%$ accuracy). We highlight that we here essentially just train a neural network with a fixed order of training examples, with recency biased rehearsal, label smoothing and forward-calibration.

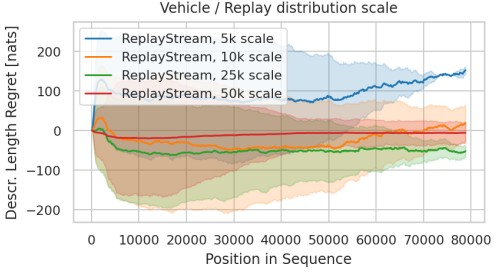 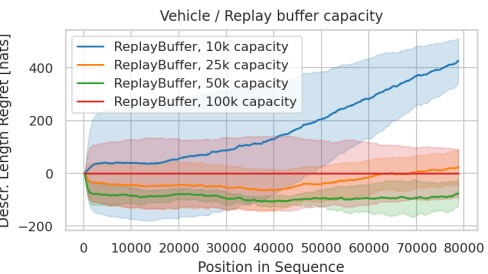

### C.7.3 CONTINUAL GEO-LOCALIZATION (CLOC)

We present results on the CLOC data introduced by Cai et al. (2021). CLOC is a large scale, chronologically sorted image classification sequence consisting of $\approx 39M$ images with their geolocation as a categorical label. For $\approx 5\%$ of the images we either received download errors, or the downloaded content could not be decoded as image data. In total we were able to obtain 37,093,769 labeled images. Figure 2 in (Cai et al., 2021) shows that the data is strongly non-stationary; and that traditional ERM training, which treats the data as i.i.d., results in a held-out error rate between 80 and 90%.

Cai et al. (2021) reserve the first 2M (5%) datapoints for pre-training the model statically before entering the online-phase, split-off $1\%$ of the data throughout for validation purposes and only evaluate the images from the first new album in each mini-batch instead of all images. Their online-learning task and metric are thus in spirit similar to the prequential MDL problem we are tackling here, in detail however sufficiently different that the quantitative results are not directly comparable. Following (Cai et al., 2021) we use a standard ResNet50 architecture and run experiments with replay buffers and replay streams. The hyperparameter sampling space is detailed in Appendix B.5. Note that we use a batch size of 128 because small batch-sizes allow the model to more quickly adapt to changes in the data distribution; and a relatively low number of 10 to 25 replay-steps / replay-streams to limit the total computational cost due to the size of the data.

We observe that heavily recency biased replay leads to significantly improved results. For replay streams we experimented with uniform replay, replay with exponentially decaying priority and with replay governed by a heavy tailed Pareto distribution (see C.6). From these the heavy tailed Pareto replay leads to the shortest description lengths. Even more than in the i.i.d. case, forward-calibration plays an crucial role to obtain good results.

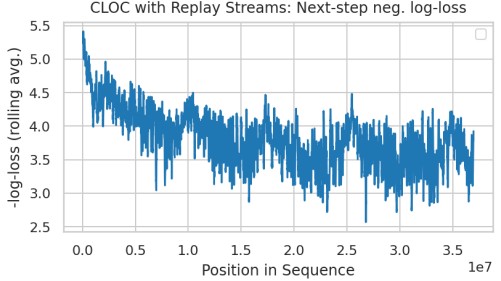 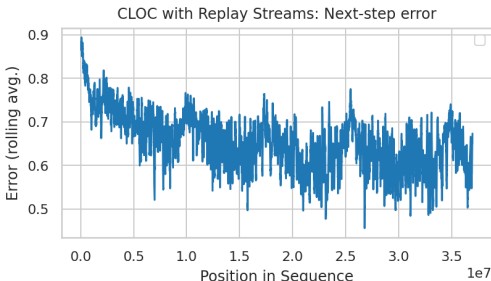

**Next-step performance** We plot the 5k rolling average next-step performance of the best performing model from our random hyperparameter sweep ($K{=}22$ Pareto distributed replay streams with a scale of 180). The model obtains a description length of 1.33G nats and a next-step error-rate of typically $60 \pm 10\%$.

**Ablations** **Left:** We show the description length regret for the best replay-buffer training run relative to the best replay-stream training run. Both experiments consumed about $5 \times 10^{18}$ FLOPs. **Right:** Regret for replay-streams without forward calibration ralative to replay streams with forward calibration:

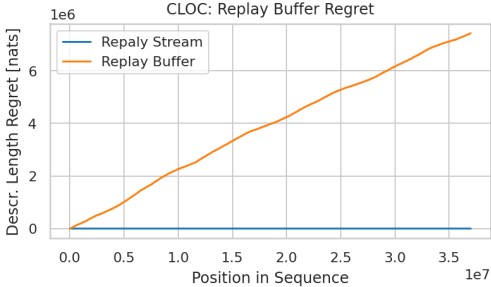 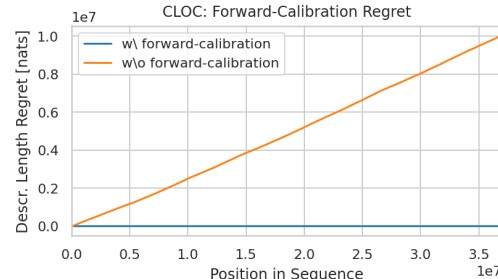

