# OpenReview forum: "Sequential Learning of Neural Networks for Prequential MDL"
_ICLR.cc/2023/Conference — ICLR 2023 poster_

### Official Review · Reviewer_zGnF · 2022-10-24

**Confidence:** 3
**Correctness:** 4
**Technical Novelty And Significance:** 2
**Empirical Novelty And Significance:** 3
**Recommendation:** 6

**Clarity, Quality, Novelty And Reproducibility:**

Clarity:
I really like the writing and the presentation of the paper. The Pareto front figures are very clear and informative. I also have a few suggestions, and please check the minor comments below.

Quality:
The paper shows extensive experimental results on different datasets using various network architectures. The empirical results are of high quality and could be very useful to future research.

Novelty:
As discussed in the strengths and weaknesses part, the novelty of the paper is limited. The experimental results are convincing but non-surprising. Overall I think it's useful empirical work for MDL objective training, which could be critical for applying machine learning models to more practical scenarios.

Reproducibility:
The paper contains many details about the experiment settings and the reproducibility seems solid.


Minor Comments:
1. "Continuous learning" and "continual learning" are both used, and I think they correspond to the same thing. "Continual learning" is probably a more widely used term.

2. The regret is not precisely defined. Given multiple forms of regret are commonly used in online learning literature, I suggest the authors provide a precise definition of the regret used in the paper.

**Strength And Weaknesses:**

Strengths:
1. The paper studies the problem of MDL objective training. This setting addresses many problems in practice where assumptions for ERM objectives are not valid.

2. The paper draws the connection between MDL objective training and ERM objective training under continual learning settings.

3. The paper conducts extensive experiments and compares several methods training under the MDL objective. The paper also proposes a practical method to achieve approximate-random in rehearsal training.

4. The paper gives a nice overview of MDL in the appendix.


Weaknesses:
1. The novelty and algorithmic contributions of the paper are limited. The proposed/studied methods are all existing approaches or simple modifications of existing ones. The paper also lacks theoretical justifications.

2. As mentioned in the paper, the experiments do not account for the learning rate annealing, which is a widely used technique for ERM training and significantly affects the final performance.


**Summary Of The Paper:**

The paper studies the problem of training deep neural networks with the minimum description length objective. The paper connects the MDL objective training to the ERM objective training but under the settings of continual learning. The paper conducts extensive experiments and compares several methods to achieve the best MDL performance with FLOP cost constraints.

**Summary Of The Review:**

The paper studies a practically very important problem, conducts extensive experiments, and provides convincing arguments to understand the experimental observations. The paper's presentation is very clear. The paper is weak on the methodology and novelty side, but still very useful given its strong and solid empirical results.

---

> ### Author Response · Authors · 2022-11-20
> **Author's reply**
>
>
> Thank you for your review and the thoughtful comments. We have incorporated all your suggestions and updated the paper accordingly – we are convinced that this improved the manuscript.
>
> We generally agree with your comments regarding strengths and weaknesses of our work. Even without introducing major theoretical or technical innovation, we believe it very valuable to empirically show that there are significantly better techniques available than the "chunk-incremental/from-scratch" approach, which is currently widely used [1, 2, 3, 4, 5, 6].
>
> Regarding theoretical justifications: (prequential) MDL itself is well motivated by a large body of literature. The individual techniques used here to improve the MDL scores on the other hand are indeed neither well understood, nor well justified. This seems analogous to the situation with SGD-based optimization of neural networks in general: The theory to understand and justify techniques often unfortunately lags behind empirical improvements.
>
> [1] Blier & Ollivier, 2018
> [2] Whitney et al., 2020
> [3] Elena Voita and Ivan Titov, 2020
> [4] Jin et al., 2021
> [5] Bornscheinet al., 2021
> [6] Perez et al., 2021

---

> > ### Comment · Reviewer_zGnF · 2022-12-12
> > **After Rebuttal**
> >
> > Thanks for the replies. I have checked the reviews of other reviewers and the corresponding responses. Overall I think the paper has some limitations but overall provide many useful empirical results. Therefore, I keep my score.

---

### Official Review · Reviewer_LsM9 · 2022-10-24

**Confidence:** 2
**Correctness:** 3
**Technical Novelty And Significance:** 3
**Empirical Novelty And Significance:** 3
**Recommendation:** 6

**Clarity, Quality, Novelty And Reproducibility:**

Clarity:

The paper is well-written and self-contained, with nice introduction to MDL in Appendix with some caveats:

- The cumulative loss is lower with the proposed method but it is not clear to me how this is the right metric to evaluate the model: if the dataset is a classification dataset, what matters is rather the test accuracy? If so, the model does less good than ERM and one of the two MDL baselines. Could you clarify this?

Quality:
- The experimental protocol is thorough, with potentially missing baselines (see above).

Novelty:
- To the best of my knowledge, applying and extensively evaluating continual learning techniques to Prequential MDL is new, as well as the proposed techniques.

Reproducibility:
- No code is provided but the Appendix contain many experimental details.

**Strength And Weaknesses:**

Strength:
- Finding alternatives to ERM, whose central hypothesis is that data is i.i.d. is an important problem to apply machine learning in many interesting settings. This work investigates a potential candidate, namely prequential MDL.
- The paper is very well-written and motivated.
- The experimental protocol seems thorough: many different architectures and datasets are considered, and ablations are done.
- Experimental results are promising: the proposed method performs better than other MDL baselines, and rival with ERM without learning schedule.

Weakness:
- It seems to me that some baselines are missing: it could be worth evaluating the test accuracy for MDL baselines on ImageNet, and mentioning the results of ERM with a learning schedule (I am surprised by the discrepancy of ERM results with/without learning schedule).
- Although MNIST, CIFAR and ImageNet are iid, wouldn't it be possible to make it non-iid, train on a smaller number of epochs and see how your method performs vs. ERM? I think it may be a more appropriate setting to use continual learning.

Minor:
- Typo: negaive (last line before Figure 4).

**Summary Of The Paper:**

This work extensively investigates continuous learning based methods for prequential minimum description length.

More precisely, the authors train MDL models using online learning with rehearsal, and propose two techniques for improving the results.
- Forward-calibration, to optimize a calibration parameter $\beta$ by using each new batch of samples for evaluation, then single gradient step on $\beta$, then training the other parameters $\theta$, before being placed in a replay buffer.
- Replay streams, to avoid implementing large replay buffers. Replay streams rely on the data stored in its original order, and maintain K pointers to some positions in the ordered data. When the k-th replay stream is called, it yield the corresponding sample and increments the pointer by one. The authors also propose a mechanism to randomly reset the pointers depending on what's needed.

The authors extensively evaluate their method compared to classical prequential MDL (models trained from scratch or fine-tuned on increasingly larger chunks of data) and compare to ERM on various datasets and neural network architectures (without learning rate schedule). They show their method to perform reasonably well against ERM given the same computational budget and better than previous baselines for prequential MDL, and perform some ablations on techniques such as forward calibration, label smoothing and weight standardization.

**Summary Of The Review:**

This work investigates an interesting alternative to ERM. It is well-written and the experimental protocol is thorough, although some baseline or experiments may be in my opinion missing to convince that the proposed method is very convincing. I wouldn't recommend acceptance for now but would increase my score if my concerns (add baselines or explain why they are not appropriate, meaning of the cumulative loss w.r.t. test accuracy) are answered.

---

> ### Author Response · Authors · 2022-11-17
> **Author's reply**
>
> Thank you for the review and taking the time to understand and consider our work.
>
> We hope the replies below address concerns raised -- but please let us know if there are further questions:
>
> ### Additional Baselines
>
> We report additional baseline metrics after running the hyperparameter-sweep outlined in the supplement.
>
> ResNet-50 model (standard width) on ImageNet
>
> - Replay Streams: pMDL score 2.21M
> - Replay Buffer (500k capacity): pMDL score 2.26M nats
> - Chunkwise-from-scratch (16 split points, <1e19 FLOPs): pMDL score 2.32M nats,
>
> Note that for chunkwise-from-scratch evaluation we use the techniques from [Bornschein, 2020], who report 3.32M nats. Our hyperparameter search thus found a significantly better result for their method.
>
> Using a 4x wider architecture:
>
> - Replay Streams: pMDL score 2.18M nats
> - Replay Buffer (500k capacity): pMDL score 2.22M nats
> - Chunkwise-from-scratch (16 split points, ~ 1e19 FLOPs): pMDL score 2.18M nats,
>
> We emphasize that we use random hyperparameter searches for all methods under consideration: Careful tuning can improve the absolute results for all methods a bit (see ResNetˆdagger in Tables 1 for example). To maintain a fair comparison and to avoid tuning some methods more than others, we chose to use random hyperparameter sweeps. Our ERM results are therefore slightly weaker than manually tuned results in the literature. However, we confirmed that our ResNet-50 implementation, when trained with the standard learning rate schedule from [He et. al. 2016], obtains the expected 79% heldout-error rate.
>
> ### ...it is not clear to me how [MDL] is the right metric to evaluate the model.
>
> Please see the separate comment to all reviewers where we lay out the reasons for evaluating with i.i.d. data. TL;DR: evaluating on i.i.d. is the hardest scenario for a continual pMDL learner and ERM trained models provide a strong baseline in terms of held-out performance as they have been tuned for decades. We do not suggest that MDL should be preferred when all the assumptions of ERM are fulfilled. However, there are situations in which one is interested in MDLs properties such as Occam's Razor or robustness to train/test contamination instead. For example [1, 2, 3, 4, 5].
>
> We will try to motivate our choices better in the upcoming revision of the manuscript.
>
>
> - [1] Causal direction of data collection matters: Implications of causal and anticausal learning for NLP; Jin et. al. 2021
> - [2] Evaluating representations by the complexity of learning low-loss predictors; Whitney et. al., 2020
> - [3] Rissanen data analysis:  Examining  dataset characteristics via description length; Perez et. al. 2021
> - [4] Prequential MDL for Causal Structure Learning with Neural Networks, Bornschein et. al., 2021
> - [5] Information-Theoretic Probing with Minimum Description Length, Elena Voita, Ivan Titov, 2020

---

> > ### Author Response · Authors · 2022-12-11
> > **Author's reply**
> >
> > Dear Reviewer,
> >
> > We believe our reply and the updated revision of the paper address the issues raised during the review -- especially regarding additional experiments on i.i.d. and non-stationary data. If you have any further questions, please don't hesitate to reach out. We would be happy to provide any additional information or clarification.
> >
> > Thank you again for your time and consideration.

---

> > > ### Comment · Reviewer_LsM9 · 2022-12-12
> > > **Rebuttal seen**
> > >
> > > Thank your for the clarifications, I think they make the submission stronger. I will raise my score since my own concerns were answered

---

### Official Review · Reviewer_zftM · 2022-10-28

**Confidence:** 3
**Correctness:** 2
**Technical Novelty And Significance:** 2
**Empirical Novelty And Significance:** 3
**Recommendation:** 6

**Clarity, Quality, Novelty And Reproducibility:**

**Clarity, Quality, Novelty:**
- I think the writing of the paper can be improved significantly in terms of succintness and clarity. The results are hard to interpret and notations are not clearly described.
- The work utilizes ideas from related fields in a novel manner.

**Strength And Weaknesses:**

**Strengths:**
- The experiments compare several methods across multiple datasets and networks.
- The paper does multiple ablation studies to better understand different components of the algorithm, although this study is relegated to the appendix.

**Weaknesses:**
1. The idea of replay streams has been well studied in the Reinforcement Learning literature but has not been cited in this work.
2. The organization of the paper can be improved. Equations are introduced without defining all of the notations. I would recommend adding a preliminaries section where notations and prerequisites are first defined.
3. The MDL principle and prequential description lengths are properly introduced only in the appendix. I would recommend a short version of this be introduced in the main text.
4. Under $Chunk Incremental/Continual Fine-tuning (CI/CF)$, a claim is made "However, recent research suggests that..." - this statement does not have a reference. Also, following it, the authors state that they run an ablation study but do not state where this ablation study is, nor the results of it.
5. Under *mini-batch incremental/replay streams* they make an assumption "We instead assume that the data is stored in its original order in permananet storage and we maintain K replay streams..." - this is an extremely strong assumption and cannot hope to hold in practice. The authors emphasize that this framework is for the setting where the i.i.d assumption fails, typically in streaming settings and therefore, I feel this assumption cannot hold in practice.
6. Interpreration of Results:
    - I am not sure what to take away from Table 1. It seems to me like ERM outperforms MI/RS in most settings which makes sense since these settings are artificially created from the i.i.d setting.
    - In Figure 1, the scale of the loss as well as errors is different for ERM compared to Prequential MDL training. THerefore, I am unsure how to compare these two. It could be argued that the difference in the plots is purely due to scale since the absolutely differences is what we compare.

**Summary Of The Paper:**

The paper proposes two new techniques: *forward-calibration* and *replay streams* for efficiently computing prequential description lengths. It particularly focuses on the computational costs associated with these methods and claims to show that they improve upon previously reported results by large margins.

**Summary Of The Review:**

~The paper in its current state is hard to read and it is not clear that the improvements are significant. Therefore, I cannot recommend acceptance. However, if changes are made and my questions are clarified, I would be willing to change my mind.~

Since the authors have improved their presentation and clarified several of my concerns, I am increasing my score to 6.

---

> ### Author Response · Authors · 2022-11-17
> **Author's response**
>
> Thank you for your time and effort understanding our work and for asking highly relevant questions. We are currently updating the manuscript and will upload a new version soon. We believe your comments and questions have helped us improve the manuscript.
>
> Below we address the questions and concerns raised:
>
> ### Replay Streams not novel:
>
> There is a sizable body of literature on the benefits of (in-memory) replay buffers and how to work around their size limitations by selecting core-sets, or reservoir-sampling, or compressed storage – we cite highlights of those lines of work. We are not aware of any proposal for sequential only access to replay data to accommodate streaming from HDDs and thus, in practice, to circumvent the problem altogether.  Can you point to any prior work that uses a technique like replay streams?
>
> ### Assumptions for Replay Streams extremely strong and cannot hope to hold in practice:
>
> On the contrary; we maintain that streaming data that can be stored on HDD but not in RAM is the default setup for deep learning these days. We admit that there are situations in which data is ephemeral and directly streamed from acquisition (sensor) to algorithm without ever being stored. But in the majority of scenarios training data is stored: ImageNet, CLOC (see appendix), web-scraped text data for large language models, data for image models like CLIP, video datasets like Kinect600: These are all datasets significantly exceeding RAM sizes;  but are stored and read from permanent storage at training speed.
>
> And large language models show that Tera- or Peta-bytes of training data can be stored and read at costs small compared to the compute resources required to perform deep learning on them.
>
> With Replay Streams we can perform sequential learning with (approximate) uniform rehearsal on all of these –which is simply impossible with in-memory replay buffers.
>
> ### Missing Reference.
>
> The reference was in the following sentence (Ash & Adams 2020); we will bring it forward to clarify the connection.
>
> ### ERM outperforms MI/RS
>
> Yes, ERM training tends to outperform pMDL training slightly, but consistently _in terms of held-out performance_, when all ERM assumptions hold. We have to emphasize that all our parameter-tuning was aimed at optimising the pMDL score. MDL is the metric of interest for this work with the reasons outlined in the introduction.
>
> We train on i.i.d. data because this is arguably the hardest scenario for a continual learner. We do not suggest that pMDL should be used when all assumptions of ERM hold. The the comparison with ERM in terms held-out performance should rather be considered a sanity check; one that indicates that additional improvements might be possible in the future. Please also see the common response to all reviewers.
>
> ### Comparison of ERM held-out error with MDL cumulative error in Figure 1
>
> You are correct, these quantities are not comparable. They can only be used for model selection within their respective training and evaluation framework: For ERM there is no clear signal to prefer the R, RG or RGB model, while MDL clearly indicates that the R model should be preferred.
>
> Table 1 shows the holdout-results when training the MNIST models either via ERM or MDL.
>
> ### Clarity
>
> We try to add a short explicit introduction to MDL after the first section instead of entirely relying on the supplement; however space is a limited resource. Can you point to some concrete notation in the paper that could use some clarification? We believe we introduce all notation before using it in the main paper. Given the limited space in the paper we are currently adding a list with notations and clarifications to the appendix.

---

> > ### Comment · Reviewer_zftM · 2022-11-19
> > **Response to Authors**
> >
> > Thank you for responding and making modifications to your manuscript.
> > 1. References regarding Replay Streams: Please take a look at "The Challenges of Continuous Self-Supervised Learning" by Purushwalkam et al. and the references therein. They discuss the use of replay buffers in related works as well.
> >
> > 2. Assumptions on Replay Streams: I'm still not convinced by your argument. I agree that imagenet or other typical vision tasks hold in this setting, but those are cases with typical i.i.d data and are not the focus of your work on sequential problems. In the streaming setting, I would expect that the amount of data flowing in is quite large and therefore it is unreasonable to store all of it. Otherwise, one could simply store all the data up to a certain time $t$ and then use typical ERM at that point, yes? Even CLIP/LAION-5B datasets would still be considered i.i.d and not non-stationary despite their massive size.
> >
> > 3. In the introduction, a lot of mathematical notation is used without introducing it formally. While I can somewhat infer what most of them mean, I would recommend adding a preliminary section to formalize some of this notation. For example, is $p_{M}(y | x, \theta$ the probability distribution over the labels that is generated by the model $M(\theta)$? What is $M_{R}$?

---

> > > ### Author Response · Authors · 2022-12-05
> > > **Author's response**
> > >
> > > Thank you for coming back to us with your comments and apologies for the late reply.
> > >
> > >
> > > ### Replay Streams not novel.
> > > We carefully read Purushwalkam et al. In their work they discuss the benefits of replay-buffers, their size limitations and various workarounds. We cite multiple of the prior works referenced therein.
> > >
> > > *We could not find any mention of a technique like replay-streams and thus maintain that it is a novel technique not studied before.*
> > >
> > > ### Assumptions on Replay Streams
> > >
> > > Our work is not focussing on the streaming setting. We study techniques for computing MDL scores, which apply to large and small, i.i.d. and non-stationary data. All recent work that uses MDL scores with Deep Learning are based on i.i.d. data [1, 2, 3, 4, 5, 6]. It is a novel contribution from our side that we use online/sequential learning techniques. We don't think we should be bound by the constraints of the online streaming community, just because we are inspired by some of their techniques to solve the MDL scoring problem.
> > >
> > > We also point out that while Purushwalkam et al. explicitly consider the streaming scenario, in practice we could use replay-streams for all their experiments: data is stored on disk but much too large to fit into RAM; which fits exactly our assumptions. For their experiments and with replay-streams, limiting rehearsal is a choice, not a technical necessity.
> > >
> > > Furthermore, we reckon that if indeed faced with a large-volume streaming problem, variations of the replay-stream technique can be devised to support large on-disk FIFO buffers with Tera- and Petabytes of capacity that approximate random rehearsal while only requiring efficient sequential read- and write access.
> > >
> > > > Otherwise, one could simply store all the data up to a certain time t and then use typical ERM at that point, yes?
> > >
> > > Using typical ERM training on data up to certain times t corresponds to the chunk-incremental / from-scratch approach, which is the current state-of-the-art for MDL estimation on i.i.d. data [1, 2, 3, 4, 5, 6]. Our experiments show convincingly that this approach is almost always suboptimal compared to the techniques we propose (please see Figures 1, and 6, and Table 1 [prior works]). This is one of our main findings: One could, but probably should not.
> > >
> > > ### Clarity
> > >
> > > We will add a preliminaries section in case the paper will be accepted.
> > >
> > > On page 2 (upper half) we write: "…where $D={(x_t, y_t)}$ is the sequence of inputs $x_t$ and associated prediction targets $y_t$; $p_M(y | x, θ)$ denotes a model family $M$ and $\hat{θ}(D)$ an associated parameter estimator given training data $D$."
> > >
> > > $M_{R}$, $M_{RG}$, $M_{RGB}$ denote the three model families under consideration in the example on page 3: the one conditioned on the R channel only, on the R and G channels, or on R, G and B channels respectively.
> > >
> > >
> > > [1] Blier & Ollivier, 2018; [2] Whitney et al., 2020; [3] Elena Voita and Ivan Titov, 2020; [4] Jin et al., 2021; [5] Bornschein et al., 2021; [6] Perez et al., 2021

---

> > > > ### Comment · Reviewer_zftM · 2022-12-08
> > > > **Response to authors (Pt 2)**
> > > >
> > > > Thank you for your clarifications.
> > > >
> > > > **Novelty of replay streams:**
> > > > Perhaps you mean that the way you populate the replay buffer is different? I would agree with that statement. Claiming that the idea has not been studied before however is strong.
> > > >
> > > > **Assumptions on Replay Streams:**
> > > > I assumed that non-stationary data would usually consider streaming as one of the applications. However, I understand that this is not the setting you are focussing on and therefore agree with your response.
> > > > That said, I am still not convinced of the importance of sequential learning while you have all of the data available. Is there a way to modify your approach to handle the streaming setting?
> > > >
> > > > I still have some concerns about the importance of the problem but the authors have largely clarified by questions and therefore, I am increasing my score.

---

### Official Review · Reviewer_7xY3 · 2022-10-31

**Confidence:** 2
**Correctness:** 3
**Technical Novelty And Significance:** 2
**Empirical Novelty And Significance:** 2
**Recommendation:** 5

**Clarity, Quality, Novelty And Reproducibility:**

This paper is well written and clearly presented.

The focus of the paper is investigating the trade-off between minimizing prequential description lengths and computational overhead. Several heuristic methods are compared empirically compared in the experiment section.

However, the significance of minimizing prequential description lengths is not sufficiently clear, which is merely discussed in the introduction section with a toy case. In this sense, it is difficult to evaluate the paper in a broader view.

The code should be made public to validate reproducibility owing to the fact that implementation details are missing in the text.

**Strength And Weaknesses:**

It makes an encouraging attempt to bridge the gap between theory and practice for minimum description length.

**Summary Of The Paper:**

This paper investigates the problem of minimizing prequential description lengths for image datasets with neural networks, primarily dealing with the issue of computational overhead. The proposed method (Mini-batch Incremental Training with Replay Streams, MI/RS) is inspired by  continual learning and its effectiveness is demonstrated empirically across traditional image classification datasets and neural network architectures.

**Summary Of The Review:**

See “Strength And Weaknesses” and “Clarity, Quality, Novelty And Reproducibility”.

---

### Author Response · Authors · 2022-11-10
**Shared response to all reviewers: Why did we evaluate on i.i.d. data?**

We thank all reviewers for their thoughtful comments and that they took the time to reflect on the potentially unfamiliar perspective on evaluation and model selection. We understand that all reviewers agree that there are no technical or theoretical issues and that we provide a thorough empirical evaluation.

A concern among some reviewers is the significance of minimizing description length given i.i.d. data. The value of MDL lies in situations in which the assumptions of ERM are violated and where test set evaluation becomes nonsensical: In Appendix D.7 we show that treating CLOC as an ERM problem results in 80-90% error rate, while we obtain 55-65% next-step error rate. Section 2 and appendix D.1 show that even given trivial model selection choices, ERM fails to identify the more appropriate model architectures (due to missing Occam's Razor). We believe the value of alternative evaluations beyond ERM is obvious in these cases.

**Then, Why did we evaluate on i.i.d problems, where MDL is admittedly not necessary?**
 * Because it is one of the hardest scenarios a sequential learner might face.
 * Because it allows us to evaluate our techniques in terms of final held-out performance against strong baselines.

We conjecture that our results would be much less insightful if we had focused our evaluation on non-stationary data: Only very few prior works compute description lengths with neural networks; all of them use the block-wise estimate because they are interested in MDL's Occam's Razor rather than its ability to handle non-stationarity. The vast literature on MDL with non-stationary data on the other hand considers only simple models (such as exponential family); scenarios that would be considered too toy-ish for neural network training => There are hardly any external baselines to compare to.

**TL;DR: We evaluate on i.i.d. data because it is the hardest thing to do and it stress tests the techniques under investigation.**

Secondly: MDL is a universal objective: Consider a practitioner aiming to create a predictive model with optimal performance on future data, but without reliable information whether it originates from a stationary or non-stationary process. Without strong sequential learners, the practitioner will first have to test whether the problem can be treated as stationary, identify potential (near) duplicates, and then consequently choose either heldout- or MDL/forward-validation as the objective.
If however we had strong sequential learners in exactly the sense that we investigate in our paper, then such a two-stage process would not be necessary and pMDL could be the default choice that obtains the best performing model for both i.i.d. and non-i.i.d data.

**TL;DR: We evaluate on i.i.d. data because obtaining competitive performance would make pMDL in practice the universal objective which it is in theory.**

---

### Author Response · Authors · 2022-11-18
**New revision**

We uploaded an updated revision of the manuscript that contains various improvements suggested by the reviewers. We believe that this addresses the majority of the concerns raised with the paper:

* Appendix C.7 now contains additional experiments on non-stationary data. In particular we ran experiments on Split-CIFAR10 and on a vehicle classification data sequence. The experiments show that the proposed techniques improve results on the non-stationary settings throughout.
The regret plots clearly illustrate how the scale of the replay-distribution (or the size of the replay-buffer respectively) influences the learners ability to quickly adapt to new data and to maintain optimal performance on data from old tasks.

* Appendix B.4 contains additional results on ImageNet requested by Reviewer LsM9.
* We added an additional discussion of the properties of pMDL based model selection to Appendix A.4.
* We added the definition of regret in the main paper; and explicitly introduce the notation in the Appendix where we summarize the mathematical framework behind MDL.
* Fixed various minor typos; including the ones pointed out by the reviewers.
* We highlight that we will release the source code for our implementation.

---

### Decision · Program_Chairs · 2023-01-20

**Decision:**

Accept: poster

**Justification For Why Not Higher Score:**

The paper was initially rated as borderline, given the limited novelty and technical contributions. The authors' responses clarified some of the concerns of the reviewers. The reviewers are now leaning toward accepting the work. The AC, however, does not see even stronger support from the reviewers to justify a higher score.

**Justification For Why Not Lower Score:**

The reviewers increased their scores after the authors' responses.

**Metareview: Summary, Strengths And Weaknesses:**

The paper presents two methods, (1) forward calibration and (2) replay streams, for computing prequential description lengths in continuous learning-based methods.

Main strength:
+ Thorough experimental validation. Promising results. The paper shows results better than other MDL baselines.
+ Clearly written paper.

Weakness:
- Novelty and algorithmic contributions are limited. These methods are well-studied in the Reinforcement Learning literature but not cited.

The paper initially received borderline scores (including two marginally above the acceptance threshold and two marginally above the acceptance threshold). After the authors’ responses, the reviewers’ concerns are largely alleviated.
- “… largely clarified by questions and therefore, I am increasing my score” (Reviewer zftM)
- “Overall I think the paper has some limitations but overall provides many useful empirical results.” (Reviewer zGnF)
- “Thank your for the clarifications, I think they make the submission stronger. I will raise my score since my own concerns were answered” (Reviewer LsM9)

While the paper was initially rated as borderline, the reviewers are now all leaning positively toward this work. The AC thus recommends accepting the paper.


**Note From Pc:**

if the above contains the word "oral" or "spotlight" please see: "oral" presentation means -> notable-top-5% and "spotlight" means -> notable-top-25%. As stated in our emails, we are disassociating presentation type from AC recommendations